

# Association of military-specific reaction time performance with physical fitness and visual skills

Danica Janicijevic[1,2,3], Sergio Miras-Moreno[4], Alejandro Pérez Castilla[4], Jesus Vera[5], Beatriz Redondo[5], Raimundo Jiménez[5] and Amador García-Ramos[4,6]

[1] Faculty of Sports Science, Ningbo University, Ningbo, China
[2] Research Academy of Human Biomechanics, The Affiliated Hospital of Medical School of Ningbo University, Ningbo University, Ningbo, China
[3] University of Belgrade, Faculty of Sport and Physical Education, The Research Centre, Belgrade, Serbia
[4] Department of Physical Education and Sport, Faculty of Sport Sciences, University of Granada, Granada, Spain
[5] CLARO (Clinical and Laboratory Applications of Research in Optometry) Research Group, Department of Optics, Faculty of Sciences, University of Granada, Granada, Spain
[6] Department of Sports Sciences and Physical Conditioning, Faculty of Education, Universidad Católica de la Santísima Concepción, Concepción, Chile

Corresponding author
Danica Janicijevic,
danica.janicijevic@fsfv.bg.ac.rs

## ABSTRACT

**Background:** The aim of the present study was to explore whether military-specific reaction time (RT) test performance is affected by individuals' physical and visual skills.

**Method:** In a single testing session, the military-specific Simple and Go, No-Go RT, aerobic power (20-m Multistage Shuttle Run test), maximal upper- and lower-body mechanical capacities (bench press and squat against different loads), and visual skills (multiple object tracking and dynamic visual acuity) of 30 young men (15 active-duty military personnel and 15 sport science students) were evaluated.

**Results:** The main findings revealed that the Simple RT and Go, No-Go RT presented (1) with aerobic power non-significant small correlations in military personnel ($r = -0.39$ and $-0.35$, respectively) and non-significant negligible correlations in sport science students ($r = -0.10$ and $0.06$, respectively), (2) inconsistent and generally non-significant correlations with the maximal mechanical capacities of the upper- and lower-body muscles ($r$ range $= -0.10$, $0.67$ and $-0.27$, $0.48$, respectively), (3) non-significant correlations with visual skills ($r$ magnitude $\geq 0.58$) with the only exception of the Go, No-Go RT that was significantly correlated to all visual variables in the group of students (*i.e.*, students who achieved better results during visual tests had shorter RT; $r$ magnitude $\geq 0.58$), and (4) none of the physical and visual variables significantly predicted the Simple RT or Go, No-Go RT.

**Conclusion:** Altogether, these results indicate that military-specific RT performance is generally independent of physical and visual skills in both military personnel and active university students.

## INTRODUCTION

Reaction time (RT) tests have been regularly used for assessing the rapidness of the central nervous system to perceive and process sensory stimuli and to produce a relevant motor response. The list of tests that have been used for assessing RT in special populations (*i.e.*, athletes, military personnel, police officers, *etc.*) is very large (*Armstrong et al., 2013*; *Gutiérrez-Dávila et al., 2013*; *Janicijevic & Garcia-Ramos, 2022*; *Jones, 2013*; *Milic et al., 2020*; *Mudric et al., 2015, 2020*). However, a common limitation of most RT tests is that they present a low specificity in terms that the RT outcomes do not reveal the individuals' capacity to respond rapidly in their specific professional situations (*Janicijevic & Garcia-Ramos, 2022*). A finding that corroborates the importance of using specific RT tests are the negligible correlations ($r$ range = −0.071 to 0.022) found between the recently validated military-specific RT test and a standard no military-specific (*i.e.*, computer-based) RT test (*Janicijevic et al., 2021*). This result highlights that different RT test modalities do not share a significant amount of common variance. It is also known that the duration of the RT depends on non-modifiable (*e.g.*, gender and age) and modifiable (*e.g.*, personality and experience with the task) factors (*Lange et al., 2018*; *Shelton & Kumar, 2010*). However, there is no clear consensus about whether the RT is affected by someone's physical and visual skills, and if those factors equally affect different RT test modalities.

Regular aerobic exercise is known to have a number of physiological benefits such as improving cardiorespiratory function, body composition, or muscular endurance (*Blair et al., 1989*; *Swain & Franklin, 2006*). However, it is less evident whether aerobic exercise can also positively affect cognitive functions (*Garg et al., 2013*), which could be expected due to the increased mitochondrial biogenesis (*Steiner et al., 2011*) and higher cerebral blood flow (*Kleinloog et al., 2019*). These factors could be also responsible for the higher efficiency of nutrients and oxygen delivery to the brain in individuals who regularly practice aerobic exercise (*Vogiatzis et al., 2011*). The positive effect of aerobic exercise on cognitive function can be also manifested by the significant and negative correlations found in previous studies between aerobic power and RT performance (*Gentier et al., 2013*; *Huang et al., 2015*; *Reigal et al., 2019*; *Shivalingaiah, Vernekar & Naik, 2018*; *Westfall et al., 2018*). However, all these significant associations were reported during tests that were non-specific to the activities commonly performed by the populations in which they were administered (*i.e.*, computer-based RT tests) (*Gentier et al., 2013*; *Huang et al., 2015*; *Shivalingaiah, Vernekar & Naik, 2018*; *Westfall et al., 2018*) or represented a sum of the RT and movement time (*Reigal et al., 2019*). Therefore, it would be of interest to explore whether aerobic power is also significantly associated with outcomes of specific RT tests.

The associations between RT and the maximal mechanical capacities of the muscles are fairly inconsistent and generally low in healthy adults (*Clarke & Glines, 2015*; *Faulkner et al., 2007*; *Hodgkins, 2013*; *Smith, 2013*), although those associations are reported to be significant and important predictors of falling incidences in the elderly (*Faulkner et al.,*

*2007*; *Jiménez-García et al., 2021*; *Lord & Castell, 1994*). Furthermore, the correlations between the velocity of the movement (usually expressed as the movement time needed to complete a certain RT task) and RT range from negligible (*Clarke & Glines, 2015*; *Hodgkins, 2013*; *Smith, 2013*) to moderate (*Pierson, 2013*). To our knowledge, only one study has explored the associations between muscle power and RT in athletes demonstrating low but significant negative correlations (*Dane, Hazar & Tan, 2008*). Worth mentioning is also the study of (*Clarke & Glines, 2015*) who explored the association between RT and a number of different anthropometric, maturity, strength, and performance variables and interestingly none of them was significantly associated with the RT. Due to the generally inconclusive findings for the correlations between mechanical variables and RT in healthy young participants, more studies are needed to shed more light on this topic.

Previous studies have suggested an association between different visual skills and RT. *Scott, Feuer & Jacko (2002)* observed that visual acuity and color vision defects were associated with computer task speed in patients with age-related macular degeneration. Additionally, RT has been associated with the ability to quickly assess the position and direction of an object in space (*i.e.*, visual perception), however, no link has been reported between RT and kinetic visual acuity and visual field (*Kohmura et al., 2007*; *Mańkowska et al., 2015*). Both visual skills variables and RT appear as important predictors for estimating sport vision performance (*Kudo et al., 2021*), cognition in older women (*Anstey, Lord & Williams, 1997*), driving skills (*Plainis & Murray, 2002*), and falling prevalence (*Lord et al., 1994*) indicating their potential co-dependent nature. The relationship between visual skills and RT was neither explored in in-duty military personnel although they are required to be vigilant, fast in decision making and to have maximal concentration during each professional task (*Yanovich et al., 2015*). Also, having the short and accurate RT is of crucial importance for military personnel, especially during the combat situations when they need to respond to different stimuli. Keeping in mind the importance of having short RT in daily and professional activities of military personnel, it seems important to explore which are the variables affecting their RT duration. Therefore, the aim of the present study was to explore whether military-specific RT test performance is affected by individuals' physical and visual skills. We hypothesized that individuals with higher aerobic power and enhanced visual skills will present shorter RT, while the RT will be more strongly correlated to visual skills than aerobic power. The hypothesis regarding the associations between RT and strength performance could not be set due to the inconclusive results.

## MATERIALS AND METHODS

### Participants

Fifteen professional active-duty Spanish military personnel specialized in on-land activities (age = 28.8 ± 4.8 years, height = 178 ± 7 cm, and body mass = 76.5 ± 9.6 kg) and 15 sports science students (24.1 ± 4.6 years, height = 178 ± 7 cm, and body mass = 78.2 ± 10.0 kg) volunteered to participate in this study. In order to be included in the study, participants needed to (1) be free from chronical diseases, (2) be free from injuries in the last 3 months,

and (3) present a monocular visual acuity ≤0.0 log MAR in each eye with the best refractive correction. All the participants were physically fit (maximal oxygen consumption ($VO_2$) estimated by the 20-m Multi-stage Shuttle run test = $45 \pm 6$ ml/kg/min; bench press one-repetition maximum (1RM) = $76 \pm 14$ kg; squat 1RM = $104 \pm 22$ kg). The military personnel are one of population that places importance on RT because in combat situations their life and the life of people in their proximity may depend on the ability to respond to different stimuli as quickly and as accurately as possible, which was the main reason for including them in this study. Sports science students (*i.e.*, a homogenous non-military group) was also necessary to be included to make comparisons between dependency of RT from physical and visual variables. The participants signed an informal consent form before the study onset and the study was approved by the University of Granada Institutional Ethical Committee (2356/CEIH/2021).

## Study design

A cross-sectional study design was used to explore whether the ability to react rapidly to a visual stimulus is influenced by someone's physical fitness and visual skills. During a single testing session, participants performed three test types in the following order: (1) previously validated military-specific RT tests (Simple and Go, No-Go RT tests) (*Janicijevic et al., 2021*), (2) visual tests (Multiple object tracking and Dynamic visual acuity tests), and (3) physical fitness tests (20 m Multi-stage Shuttle Run test (for assessing aerobic power) and bench press and squat against different loads (for assessing the maximal mechanical capacities of lower- and upper-body muscles)). The pause between test types was 5 min. All measurements were performed between 8 AM and 2 PM and under similar environmental conditions.

## Procedure

Military personnel and sports science students were evaluated separately. Military personnel were evaluated in their assigned military base (Guzman "El bueno", Cordoba) and sport science students in the faculty of sport sciences (University of Granada, Granada, Spain). The single testing session was the same for all participants. Upon the entrance to the testing facilities, they performed three test types (*i.e.*, RT tests, visual tests and physical fitness tests) in a sequential order. The description of the tests is provided below:

- *Military-specific RT tests:* a previously validated RT test was used to assess RT during simulated military combat situations. The tests consisted of watching a 4-min video through virtual reality glasses and to respond to the stimuli by pressing a button of the gun-shaped mouse. The 4-min video consisted of a wood in which camouflaged military personnel were popping out behind different bushes. Two testing modalities were implemented, Simple RT (*i.e.*, military personnel always appeared pointing with the rifle towards the camera) and Go-No Go RT (*i.e.*, military personnel randomly appeared with the rifle pointing to the camera ("true" stimulus) or with their arms in the air ("false" stimulus)). Therefore, in the Simple RT participants were instructed to respond as soon as they perceived the military personnel in the video, while in the Go, No-Go RT participants

needed to react only when they perceived the "true" stimulus. The total number of stimuli was 56 in both tests, while the number of true and false stimuli was equal in Go, No-Go RT test (*i.e.*, 28).

A custom-made LabView program (National Instruments, version 8.2.1; Austin, TX, USA) was used for presenting the video and detecting the moments when participants reacted to the stimuli by pressing the button of the gun-shaped mouse. The utilization of the custom-made LabView program allowed synchronization of the initiation of the video and the moments when responses occurred. The virtual reality glasses (Oculus Quest 2; Meta Platforms, Cambridge, MA, USA) were wirelessly connected to the computer using the Virtual Desktop app (version 1.20.19) which allowed having the external control over the content presented through the virtual reality glasses. The RT was calculated as a time elapsed between the stimulus presentation (*i.e.*, moment when the rifle of the military personnel fully appeared in the video, defined using a program of slow-motion analysis) and the instant of the response occurrence (*i.e.*, moment when the button of the gun-shaped mouse was pressed).

- *Physical fitness tests:* the 20-m Multi-stage Shuttle run test was used to evaluate the maximal aerobic power (*Ramsbottom, Brewer & Williams, 1988*). The strength tests were performed before the endurance test. The strength test consisted of assessing the load-velocity relationship variables (maximal load ($L_0$) maximal velocity ($v_0$), and area under the load-velocity relationship line ($A_{line}$)) during the squat and bench press exercises. The mean velocity was recorded with a linear velocity transducer (T-Force System; Ergotech, Murcia, Spain) during an incremental loading test from 10 kg until the mean velocity of the barbell was lower than 0.60 m/s. Both exercises were performed with a free-weight barbell.

- *Visual tests*: participants performed multiple object tracking and dynamic visual acuity tests using the same 17.3-inch LCD ASUS laptop screen (VivoBook Pro 17 N705; width and height were 41.5 and 27 cm, respectively) with a resolution of 1,366 × 768 pixels. Participants were seated at 50 cm and 1 m from the screen for each task, respectively. The following tests were performed in a randomized order.

*Multiple object tracking (MOT) test* is a perceptual-cognitive task that explores multifocal attention and complex motion information (*Yantis, 1992*). The task consisted of following three out of eight balls (diameter 2.06°) that were randomly illuminated in green during 2 s, while the rest of the balls stayed black. Participants were instructed to track these three balls after they stopped being illuminated for additional 10 s. All balls moved randomly at a constant speed and following linear pattern. Balls deviated from the smooth path only when they collided against each other or the walls. Once the 10 s period ended, all the balls froze and the numbers were assigned to each ball (*i.e.*, from 1 to 8). Afterwards, participants were asked to identify the three balls that were originally illuminated based on their location in the display (*Fehd & Seiffert, 2008*). The initial speed of the balls was 26.3 degrees/s, and it decreased or increased by 0.05 log in a function of whether participant failed or guessed correctly which balls were illuminated (*Levitt, 1971*). The staircase

stopped after six reversals and the threshold was estimated by the average speed used in the last four reversals. This average speed value for the MOT task was the dependent variable.

*Dynamic visual acuity (DVA) test* was performed using the moV& dynamic visual acuity software (V&MP Vision Suite, Waterloo, Canada). The DVA was measured for random walk motion paths at 2.31 m/s (*i.e.*, 30°/s) (*Yee et al., 2021*). The optotype used was a black "Tumbling E" that was presented in a white background in four orientations (*i.e.*, branches of the letter E facing up, down, left or right). Participants needed to indicate the orientation of the branches of the letter E pressing the arrow keys of the keyboard as fast and as accurately as possible. It is important to note that the letter E could enter and exit in the screen at random locations, following a non-linear path but always at a constant speed. All dynamic visual acuities measured with moV& were logMAR size thresholds, while the speed of the letter was fixed and the size diminished as the test progressed. The participants needed to identify correctly three out of five targets of one size in order to progress to another level (*i.e.*, smaller size). The first letter size was 0.8 logMAR, and the letter sizes decreased in steps of 0.1 logMAR. The test ended when participants did not succeed to identify correctly the orientation of at least 3 out of 5 letters for a given size. The dependent variables were DVA threshold and DVA RT for the random motion paths.

## Statistical analysis

The Shapiro-Wilk test showed that all the variables were normally distributed ($p > 0.05$), except for the DVA threshold variables ($p \leq 0.05$). The associations between Simple RT and Go, No-Go RT and the fitness and visual variables was assessed through the Pearson's correlation coefficient ($r$), except for the DVA threshold and DVA RT in which the Spearman correlation coefficient was used since DVA threshold was not normally distributed and DVA RT is an ordinal variable. In order to explore the predictive power of the independent variables (*i.e.*, aerobic power, mechanical variables, and visual skills) on the Simple RT and Go, No-Go RT two multiple linear regressions were modelled applying the standard enter model. The scale used to interpret the magnitude of the correlation coefficients was the following: 0.00–0.09 trivial; 0.10–0.29 small; 0.30–0.49 moderate; 0.50–0.69 large; 0.70–0.89 very large; 0.90–0.99 nearly perfect; 1.00 perfect (*Hopkins et al., 2009*). All statistical analyses were performed using SPSS software version 20.0 (SPSS Inc., Chicago, IL, USA). Statistical significance was set at an alpha level of 0.05.

## RESULTS

Considering the whole sample, the association between the RT variables and aerobic power was negative and small ($r = -0.27$ for Simple RT and $-0.20$ for Go, No-Go RT) (Fig. 1). The subgroups analyses showed stronger correlations for the military personnel ($r = -0.39$, $-0.35$) than sport science students ($r = -0.10$, $0.06$) for the Simple RT and Go, No-Go RT respectively. The correlations never reached statistical significance in any group.

Associations between RT variables and mechanical variables ranged from small and negative to large and positive ($r$ range: $-0.27$, $0.67$) (Figs. 2 and 3). The magnitude of those correlations was non-systematically distributed, being the magnitudes of correlation coefficients similar for different L-V relationship parameters ($r$ range: $L_0 = -0.27$ to $0.43$;

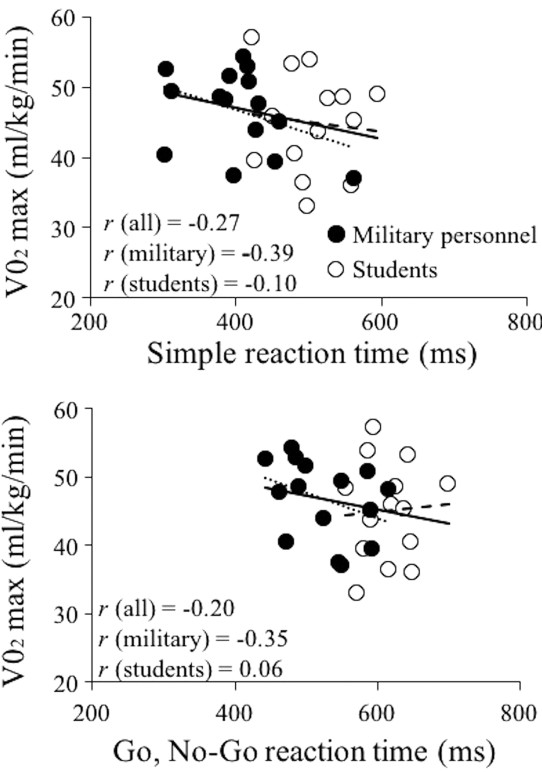

**Figure 1 Linear regression models of maximal oxygen consumption and reaction time.** Linear regression models obtained between maximal oxygen consumption ($VO_2$) max and simple reaction time (upper panel) and Go, No-Go RT (lower panel) considering the whole sample (full and empty dots and full lines), military personnel (full dots and dotted lines) and sport science students (empty dots and dashed lines). *r*, Pearson correlation coefficient.

$V_0$ = −0.18 to 0.48; Aline = −0.27 to 0.67), and exercises (*r* range: bench press = −0.17 to 0.48; squat = −0.27 to 0.67). The only significant correlation was obtained between Go, No-Go RT and Aline in military personnel (*r* = 0.67).

Associations between RT variables and visual variables (MOT, DVA threshold, and DVA RT) were ranging from −0.58 to 0.66 (Fig. 4). Although the magnitude of the correlations was non-systematically distributed, the highest correlations considering each visual variable were achieved for the Go, No-Go RT modality in the students group. Specifically, students who more successfully performed visual tests (MOT, DVA threshold and DVA RT) had also shorter RT, being the correlation coefficients large and significant for all variables (*r* magnitude always significant and higher than 0.58). None of the other correlation coefficients reached statistical significance.

Considering that we had seven predictors, that the sample size was 30, and that the α was set to 0.05, predictive power (1-β) of multiple linear regressions was 0.77, while the effect size of the study was 0.20. Multiple linear regression models were also used to test if aerobic power, load-velocity relationship variables, and visual variables can significantly predict Simple RT (first model) and Go, No-Go RT (second model). The overall regression models were not significant, explaining only ~25% and ~39% of the common variance, for the Simple RT and Go, No-Go RT, respectively. None of the variables significantly

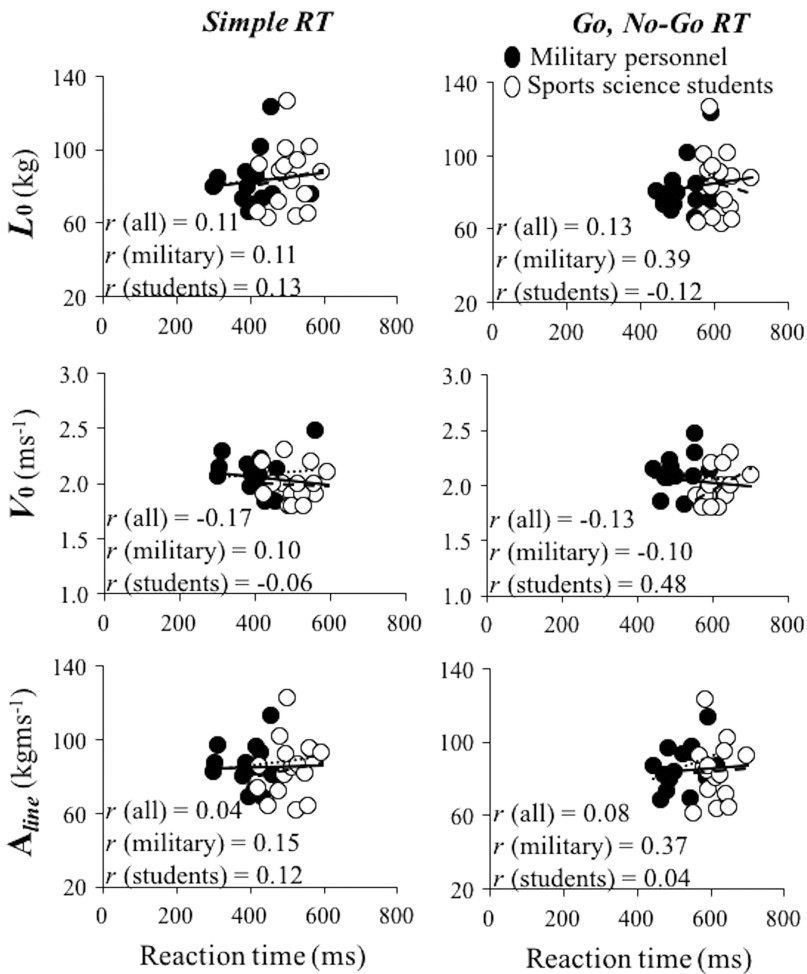

**Figure 2 Linear regression models obtained between the load-velocity relationship variables obtained during the bench press exercise and reaction time.** Linear regression models obtained between the load-velocity relationship variables obtained during the bench press exercise ($L_0$, maximal theoretical load (upper panels), $V_0$, maximal theoretical velocity (middle panels) and area under the load-velocity relationship line ($A_{line}$) (bottom panels)) and simple reaction time (left panels) and Go, No-Go RT (right panels) considering whole sample (full and empty dots and full lines), military personnel (full dots and dotted lines) and sports science students (empty dots and dashed lines). $r$, Pearson correlation coefficient.

predicted Simple RT ($p \geq 0.203$) and Go, No-Go RT ($p \geq 0.051$). Parameters of the linear regression model used for exploring predictive validity of the aerobic power, mechanical and visual variables on the Simple RT and Go, No-Go RT are depicted in the Tables 1 and 2, respectively. Considering that we had seven predictors, that the sample size was 30, and that the α was set to 0.05, predictive power (1-β) of multiple linear regressions was 0.77, while the effect size of the study was 0.20.

## DISCUSSION

The aim of the present study was to explore whether the military-specific RT test performance is affected by the individual's physical and visual skills. For this purpose, we assessed aerobic power, upper- and lower-body maximal mechanical capacities and visual

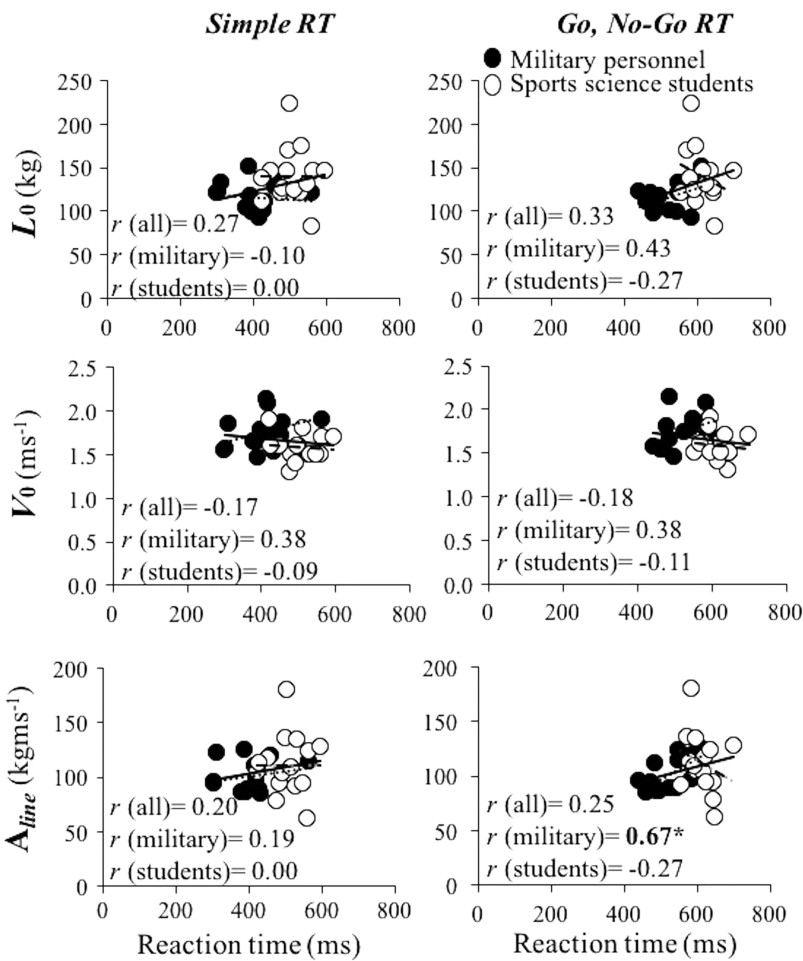

**Figure 3** **Linear regression models obtained between the load-velocity relationship variables obtained during the squat exercise and reaction time.** Linear regression models obtained between the load-velocity relationship variables obtained during the squat exercise ($L_0$, maximal theoretical load (upper panels), $V_0$, maximal theoretical velocity (middle panels) and area under the load-velocity relationship line ($A_{line}$) (bottom panels)) and simple reaction time (left panels) and Go, No-Go RT (right panels) considering whole sample (full and empty dots and full lines), military personnel (full dots and dotted lines) and sports science students (empty dots and dashed lines). r, Pearson correlation coefficient. An asterisk (*) denotes significant correlations at the level of $p \leq 0.05$.

skills of 15 military personnel and 15 sport science students. The main findings revealed that (1) aerobic power was not significantly related to RT, although the magnitude of the correlations was greater in military personnel compared to sport science students, (2) inconsistent and generally non-significant associations were found between the load-velocity relationship variables and RT, with the only exception of the significant correlation between Go, No-Go RT and $A_{line}$ in military personnel ($r = 0.67$), (3) visual skills was not significantly related to RT performance, with the only exception of Go, No-Go RT that was significantly correlated to all visual variables in the group of sport science students (*i.e.*, students who achieved better results during visual tests had shorter RT, r magnitude $\geq 0.58$) and (4) none of the physical and visual variables significantly

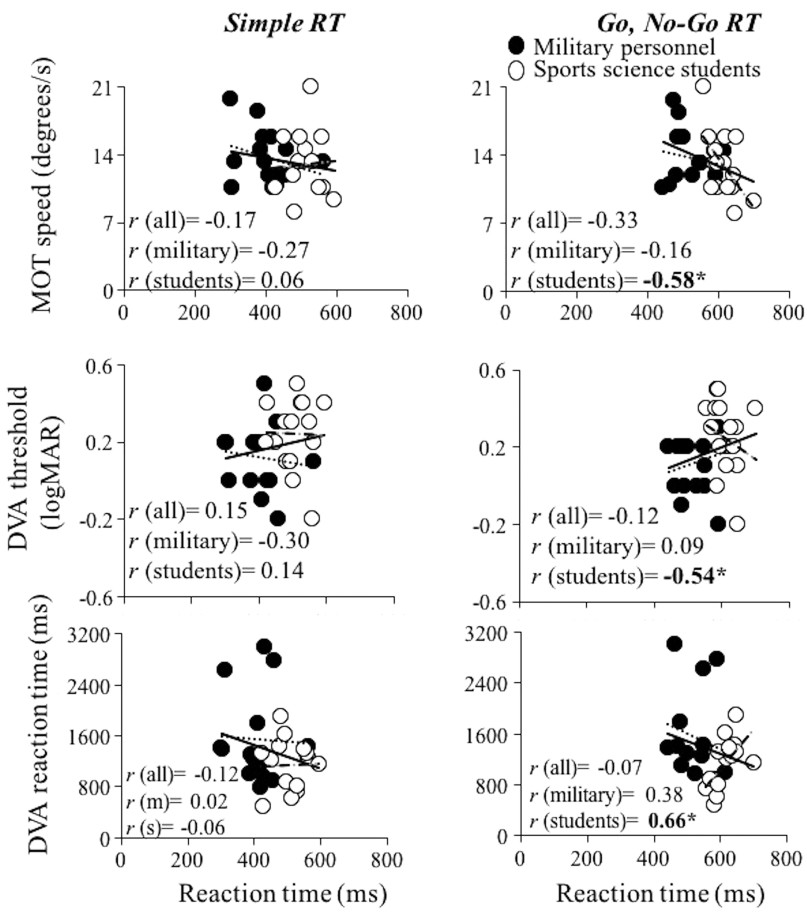

**Figure 4 Linear regression models between visual variables and reaction time.** Linear regression models obtained between the visual variables (MOT (upper panels), DVA threshold (middle panels), DVA reaction time (bottom panels)) and simple reaction time (left panels) and Go, No-Go RT (right panels) considering whole sample (full and empty dots and full lines), military personnel (full dots and dotted lines) and sport science students (empty dots and dashed lines). *r*, Pearson correlation coefficient. An asterisk (*) denotes significant correlations at the level of $p \leq 0.05$.

predicted Simple RT and Go, No-Go RT. Practically, these results indicate that RT is generally independent from physical and visual skills in healthy young males, and that neither Simple RT nor Go, No-Go RT can be predicted nor are influenced by physical and visual function. An only exception to this is Go, No-Go performance of sports science students which seems to be affected by their visual function.

Unlike in several studies (*Gentier et al., 2013*; *Huang et al., 2015*; *Reigal et al., 2019*; *Shivalingaiah, Vernekar & Naik, 2018*; *Westfall et al., 2018*), aerobic power of our participants was not significantly associated with RT performance and the magnitude of the associations was small for the whole sample or only sports science students, while moderate levels of associations were reached for the military group. Possible explanation of such a discrepancy may lie in the larger number of participants recruited in the other studies. For example, *Westfall et al. (2018)* recruited 745 participants, *Huang et al. (2015)* 493 participants, and *Reigal et al. (2019)* 119 participants. The only study that recruited a

**Table 1 Parameters of the linear regression model used for exploring predictive validity of the aerobic power, mechanical and visual variables on the simple reaction time duration.**

| Independent variables | | Unstandardised beta coefficients | Standardised beta coefficients | $p$ |
|---|---|---|---|---|
| Aerobic power | $V0_2max$ (ml/kg/min) | −4.238 | −0.347 | 0.203 |
| Bench press mechanical variables | $L_0$ (kg) | 0.546 | 0.096 | 0.977 |
| | $V_0$ (ms$^{-1}$) | 82.896 | 0.178 | 0.919 |
| | $A_{line}$ (kg·ms$^{-1}$) | −2.797 | −0.450 | 0.886 |
| Squat mechanical variables | $L_0$ (kg) | −2.506 | −0.641 | 0.759 |
| | $V_0$ (ms$^{-1}$) | −220.382 | −0.557 | 0.691 |
| | $A_{line}$ (kg·ms$^{-1}$) | 3.999 | 0.886 | 0.676 |
| Visual variables | MOT speed (degrees/s) | −9.124 | −0.348 | 0.220 |
| | DVA threshold (log MAR) | −1.423 | −0.003 | 0.994 |
| | DVA reaction time (ms) | −0.037 | −0.285 | 0.465 |

Note:
$L_0$, maximal theoretical load; $V_0$, maximal theoretical velocity; $A_{line}$, area under the load-velocity relationship; MOT, multiple object tracking, DVA, dynamic visual acuity. Alpha was set to $p \leq 0.05$.

**Table 2 Parameters of the linear regression model used for exploring predictive validity of the aerobic power, mechanical and visual variables on the Go, No-Go reaction time duration.**

| Independent variables | | Unstandardised beta coefficients | Standardised beta coefficients | $p$ |
|---|---|---|---|---|
| Aerobic power | $V0_2max$ (ml/kg/min) | −2.200 | −0.217 | 0.373 |
| Bench press mechanical variables | $L_0$ (kg) | −3.173 | −0.673 | 0.824 |
| | $V_0$ (ms$^{-1}$) | −69.475 | −0.179 | 0.910 |
| | $A_{line}$ (kg·ms$^{-1}$) | 1.273 | −0.246 | 0.931 |
| Squat mechanical variables | $L_0$ (kg) | −1.545 | −0.476 | 0.801 |
| | $V_0$ (ms$^{-1}$) | −209.405 | −0.637 | 0.616 |
| | $A_{line}$ (kg·ms$^{-1}$) | 3.637 | 0.970 | 0.614 |
| Visual variables | MOT speed (degrees/s) | −11.327 | −0.520 | 0.051 |
| | DVA threshold (log MAR) | −34.631 | −0.098 | 0.793 |
| | DVA reaction time (ms) | −0.042 | −0.395 | 0.268 |

Note:
$L_0$, maximal theoretical load; $V_0$, maximal theoretical velocity; $A_{line}$, area under the load-velocity relationship; MOT, multiple object tracking, DVA, dynamic visual acuity. Alpha was set to $p \leq 0.05$.

similar number of participants was the study of *Shivalingaiah, Vernekar & Naik (2018)* who obtained significant correlation coefficients between aerobic power and RT both in the group of runners and controls. Nevertheless, the correlation coefficient never exceeded moderate levels $r \leq 0.5$. Although having similar aerobic power (*i.e.*, 45 and 46 ml/kg/min), possible differences in the correlation coefficients obtained in our study for sport science students ($r = -0.10, 0.06$) and military personnel ($r = -0.39, -0.35$) might be explained by the specificity of the RT task (*i.e.*, military-specific RT test). It is possible that due to the nature of the RT task military personnel could benefit more from their aerobic power when performing the test, however, future studies should test this hypothesis.

Generally low and non-significant correlations were found between the load-velocity relationship variables obtained during bench press and squat exercises and RT, indicating

that these variables are fairly independent of RT in healthy young males (see Figs. 3 and 4). The findings are in line with other studies that have explored the association of a variety of strength tests and RT (*Clarke & Glines, 2015*; *Faulkner et al., 2007*; *Hodgkins, 2013*; *Smith, 2013*). However, other studies indicate that the associations between strength tests and RT are stronger as we get older (*Faulkner et al., 2007*; *Jiménez-García et al., 2021*; *Lord & Castell, 1994*). Our findings are also in line with the majority of the studies that have explored the associations between velocity capacity and RT (*Clarke & Glines, 2015*; *Hodgkins, 2013*; *Smith, 2013*), but opposite to the findings of the only study that has explored associations between muscle power and RT (*Dane, Hazar & Tan, 2008*). However, the results of (*Dane, Hazar & Tan, 2008*) should be taken with precaution since the actual test for power assessment was not described. Generally, all these findings indicate that the maximal mechanical capacities and RT do not share significant amount of common variance in healthy young male individuals.

All visual skills variables of sport science students were significantly correlated with the Go, No-Go RT, while other correlations were low and non-significant. It is possible that sport science students were leaning more on their visual searching ability while performing Go, No-Go RT than military personnel due to the specificity of the task, however, it is only a speculation. Surprisingly, multiple linear regression analyses demonstrated that neither physical nor visual variables have power to predict RT duration, explaining only ~25% and ~39% of the common variance, for the Simple RT and Go, No-Go RT, respectively. The design of this study allowed obtaining straightforward information that RT duration is not influenced by the physical and visual skills, however, there are some limitations that should be acknowledged. The sample size is lower than many similar correlational studies and the reason for that is the inability to test more military personnel due to their dense daily schedules. Also, all of our participants performed only military-specific RT tests, and not sport-specific RT tests, which impeded us from drawing clear conclusions regarding the influence of the specificity of the RT task on the predictive power of physical and visual skills.

## CONCLUSIONS

The outcomes of military-specific Simple RT and Go, No-Go RT tests cannot be predicted using physical and visual skills variables in healthy young male individuals. The correlation coefficients generally never reached statistical significance between RT and physical and visual skills variables. The only clear exception was the significant correlations between Go, No-Go RT and all visual variables in the group of sport science students. On the other hand, from the 42 coefficients of correlations obtained between physical function variables and RT, the only significant one was obtained between Go, No-Go RT and $A_{line}$. Therefore, it seems evident that RT is not influenced by physical function in these populations. Although physical and visual variables cannot be used for predicting military-specific RT performance, future studies should investigate in detail the potential effect of the RT test specificity on the visual strategies, especially in less skilled participants.

### Funding

This project was funded by the CEMIX (Centro Mixto UGR-MADOC, Army of Spain; grant reference: 5/4/20 TR-COMBATE). This work was also supported by the Ministry of Education, Science, and Technological Development of the Republic of Serbia under the grant 451-03-9/2021-14/200154. The funders had no role in study design, data collection and analysis, decision to publish, or preparation of the manuscript.

### Grant Disclosures

The following grant information was disclosed by the authors:
Centro Mixto UGR-MADOC, Army of Spain: 5/4/20 TR-COMBATE.
Ministry of Education, Science, and Technological Development of the Republic of Serbia: 451-03-9/2021-14/200154.

### Competing Interests

Jesus Vera, Amador García-Ramos and Beatriz Redondo are Academic Editors for PeerJ.

### Author Contributions

- Danica Janicijevic conceived and designed the experiments, performed the experiments, prepared figures and/or tables, authored or reviewed drafts of the article, and approved the final draft.
- Sergio Miras-Moreno performed the experiments, authored or reviewed drafts of the article, and approved the final draft.
- Alejandro Pérez Castilla performed the experiments, authored or reviewed drafts of the article, and approved the final draft.
- Jesus Vera performed the experiments, authored or reviewed drafts of the article, and approved the final draft.
- Beatriz Redondo analyzed the data, prepared figures and/or tables, authored or reviewed drafts of the article, and approved the final draft.
- Raimundo Jiménez analyzed the data, authored or reviewed drafts of the article, and approved the final draft.
- Amador García-Ramos performed the experiments, authored or reviewed drafts of the article, and approved the final draft.

### Ethics

The following information was supplied relating to ethical approvals (*i.e.*, approving body and any reference numbers):

The participants signed an informal consent form before the study onset and the study was approved by the University of Granada Institutional Ethical Committee (2356/CEIH/2021).

## Data Availability

The raw data is available in the Supplemental File.

## Supplemental Information

Supplemental information for this article can be found online at http://dx.doi.org/10.7717/peerj.14007#supplemental-information.

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
