# Peer review of "Association of military-specific reaction time performance with physical fitness and visual skills"

_PeerJ, doi:10.7717/peerj.14007_

## Round 0.1 · original submission · Major Revisions

Dear authors, congratulations, your manuscript has the potential to be published by PeerJ. However, major revisions need to be done. Please, after observing all the guidelines of the two reviewers, resubmit the manuscript to our journal for the final review. This new submission is part of the process of evaluation by PeerJ but doesn't warrant it.

Reviewer 1 ·

Basic reporting

The text is clear and not ambiguous. In general, the citations are good, but I suggest the author review the citation in lines 79-81. It is possible to find more recent authors that propose the relation between aerobic exercise and cognitive function.
In line 108 please check de form of citation “(Clarke & Glines, 2015)”
The figures present the data necessary for results comprehension, but they could have high resolution.

Experimental design

Although of the small sample, the methodology is sufficiently detailed to be replicated.

Validity of the findings

The methodology is satisfactory, but the small sample is not justifiable due to the dimension of the researched population. The sample calculation should be informed.
Due to the absence of sample calculation would be important to determine the effect size of the study.
The research limitations (like the small sample and effect size absence) should be clearly specified in the results.

Reviewer 2 ·

Basic reporting

First, I would like to congratulate the authors for the manuscript. All considerations and questions are for the purpose of improving it.

Introduction

• I suggest briefly reporting characteristics of the study population, why do we assess is important?

Experimental design

Method
• Was the sample of 15 individuals sufficient to find the associations? Please provide information on sampling power.

• Were students evaluated as well? The method does not contain this information, please insert it.

• What were the adjustment variables in the multiple linear regression?

Validity of the findings

Results
• Why didn't other regression parameters appear? Beta, standardized beta, R2?

Discussion
• What is the practical application of the study?

Additional comments

no comment'

---

## Round 0.2 · accepted · Accept

Dear authors. We are happy to inform you that your manuscript has been accepted.

Regards